# A Proteomic Analysis of Detergent-Resistant Membranes in HIV Virological Synapse: The Involvement of Vimentin in CD4 Polarization

**DOI:** 10.3390/v15061266

**Published:** 2023-05-28

**Authors:** Naoyuki Iida, Madoka Kawahara, Riku Hirota, Yoshio Shibagaki, Seisuke Hattori, Yuko Morikawa

**Affiliations:** 1School of Pharmacy, Kitasato University, Shirokane 5-9-1, Minato-ku, Tokyo 108-8641, Japan; iidan@pharm.kitasato-u.ac.jp (N.I.); shibagakiy@pharm.kitasato-u.ac.jp (Y.S.); hattorisei@gmail.com (S.H.); 2Omura Satoshi Memorial Institute and Graduate School for Infection Control, Kitasato University, Shirokane 5-9-1, Minato-ku, Tokyo 108-8641, Japan; kawahara@niid.go.jp (M.K.); rhirota@mail.biken.or.jp (R.H.)

**Keywords:** proteomics, HIV-1, virological synapse, lipid rafts, DRM, vimentin

## Abstract

The cell–cell contact between HIV-1-infected and uninfected cells forms a virological synapse (VS) to allow for efficient HIV-1 transmission. Not only are HIV-1 components polarized and accumulate at cell–cell interfaces, but viral receptors and lipid raft markers are also. To better understand the nature of the HIV-1 VS, detergent-resistant membrane (DRM) fractions were isolated from an infected–uninfected cell coculture and compared to those from non-coculture samples using 2D fluorescence difference gel electrophoresis. Mass spectrometry revealed that ATP-related enzymes (ATP synthase subunit and vacuolar-type proton ATPase), protein translation factors (eukaryotic initiation factor 4A and mitochondrial elongation factor Tu), protein quality-control-related factors (protein disulfide isomerase A3 and 26S protease regulatory subunit), charged multivesicular body protein 4B, and vimentin were recruited to the VS. Membrane flotation centrifugation of the DRM fractions and confocal microscopy confirmed these findings. We further explored how vimentin contributes to the HIV-1 VS and found that vimentin supports HIV-1 transmission through the recruitment of CD4 to the cell–cell interface. Since many of the molecules identified in this study have previously been suggested to be involved in HIV-1 infection, we suggest that a 2D difference gel analysis of DRM-associated proteins may reveal the molecules that play crucial roles in HIV-1 cell–cell transmission.

## 1. Introduction

HIV-1 spreads via two modes: through the release of cell-free viruses or cell-to-cell transmission [1]. The latter is an orders of magnitude more efficient mode of viral dissemination than cell-free viruses. Cell-to-cell transmission has been observed during cell contact between HIV-1-infected/loaded cells and uninfected cells [2,3,4,5,6,7]. This cell contact induces the accumulation not only of HIV-1 components, but also their receptor CD4, coreceptors CXCR4/CCR5, and integrins at the cell interfaces to form a supramolecular structure [2,8,9]. This structure resembles supramolecular-activating clusters (SMAC) in the immunological synapse, thereby called the virological synapse (VS) [10,11,12]. It is plausible and even likely that the accumulation of these host factors contributes to the cell-to-cell transmission of HIV-1 by ensuring stable cell–cell contact and the dense conditions of the receptor/coreceptor at the contact site. In fact, all anti-CD4, anti-CXCR4, and anti-LFA1 integrin antibodies inhibit the formation of the HIV-1 VS, thus preventing HIV-1 transmission [2,9,13]. Recent studies have further revealed that, similar to the immunological synapse, mitochondria are polarized toward the HIV-1 VS in a calcium-dependent manner [14,15]. ATP synthase has also been observed to accumulate in the VS [16], suggesting that ATP synthesis is required for HIV-1 transmission and perhaps for cytoskeletal remodeling. Thus, the molecules accumulated at the site of the HIV-1 VS may play a crucial role in HIV-1 cell-to-cell transmission.

“Lipid rafts” are highly dynamic microdomains of the plasma membrane enriched in cholesterol, sphingolipids, and phospholipids [17]. Owing to the enrichment of sphingolipids and cholesterol, lipid rafts are isolated as detergent-resistant membranes (DRM) at 4 °C [18]. These microdomains serve as a platform for protein sorting, signaling transduction, and cell adhesion [17]. A proteomic study of DRM-associated proteins identified a large number of molecules involved in signal transduction (e.g., tyrosine kinases and small GTPases) [19]. In lymphocytes, these molecules are further recruited to or accumulate in the DRM upon T cell activation [20,21,22], indicating that the DRM functions as a meeting places for cell-signaling molecules, such as the T cell receptor (TCR), in the immunological synapse. Interestingly, in the HIV-1 VS, analogous to the immunological synapse, studies have shown that lipid raft markers accumulate at the cell–cell interface where HIV-1 Gag and Env proteins accumulate, and that a disruption of the DRM impairs the VS formation [23]. The HIV-1 Gag protein is primarily associated with the DRM and then forms virus particles via multimerization. The Env protein is incorporated into these virus particles through the Gag–Env interactions during particle assembly. Many studies have shown that HIV particle assembly and budding take place at DRM microdomains, suggesting that lipid rafts serve as platforms for viral component assembly [24,25,26]. The CD4 receptor is also associated with the DRM and is essential for HIV-1 entry [27,28]. Although the association of CD4 with the DRM is unnecessary for HIV-1 infection [29], cholesterol depletion impairs HIV-1 entry [30]. These data suggest that the DRM contributes to both HIV-1 entry and budding. Tetraspanin-enriched microdomains (TEM), another membrane domain, have been shown to be enriched at the sites of particle assembly [26,31] and the HIV-1 VS [26,32,33].

To gain insight into the molecular mechanisms of HIV-1 cell-to-cell transmission, Len et al. performed a quantitative phosphoproteomic analysis of the HIV-1 VS and observed a significant activation of the TCR signaling pathways, similar to the immunological synapse, despite the absence of a relevant antigen presentation [34]. Since DRM microdomains are likely involved in HIV-1 VS formation, including both stages of HIV-1 entry and release, a proteomic analysis of the DRM from cell cocultures forming the HIV-1 VS would provide additional clues for a better understanding of the molecular mechanisms of VS formation. In the present study, we isolated the DRM fractions from HIV-1-infected and uninfected cell cocultures with cell contact and compared the protein profiles to those of non-cocultured cell mixtures, using two-dimensional fluorescence difference gel electrophoresis (2D DIGE), since it is widely used and considered to be accurate and reproducible [35]. Our pilot study revealed that some of the molecules differentially enriched in the DRM fractions upon coculture were host factors that have previously been shown to be involved in HIV-1 infection and VS formation, suggesting that DRM proteomics is a valuable tool for better understanding the nature of the VS. Furthermore, we selected vimentin from the molecules identified by our analysis and explored its involvement in HIV-1 cell-to-cell transmission.

## 2. Materials and Methods

### 2.1. Cell Culture and viral Infection

Jurkat cells (human T cell line) were grown in RPMI1640 medium supplemented with 10% fetal bovine serum (FBS) at 37 °C under a 5% CO_2_ concentration. HIV-1 was produced via a transfection of the pNL43 plasmid (AF324493.2) to Lenti-X 293T cells (Clontech, Takara Bio, Tokyo, Japan) and harvested 2 days post-transfection. The Jurkat cells were infected with HIV-1 (the NL43 strain) at a multiplicity of infection (MOI) of 3 with 4 μg/mL polybrene. Then, they were cultured with fresh growth medium at 37 °C for 2 days. For the HIV-1 VS formation, the HIV-1-infected Jurkat cells were incubated with an equal number of uninfected Jurkat cells at 37 °C for 1 h.

For CRISPR-Cas9 systems, gRNAs (targeting nucleotides 849–871 of vimentin ORF, selected using CRISPRdirect) were cloned into lenti-CRISPRv2 containing a Cas9 expression cassette (Addgene, Watertown, MA, USA). The Lenti-X 293T cells were cotransfected with lenti-CRISPRv2, psPAX2 (Addgene), and pHCMV g to produce recombinant lentivirus. Stably expressing cell lines were established with puromycin selection. Vimentin-deficient Jurkat and control Jurkat (expressing Cas9 alone) were infected with HIV-1. In some experiments, 1.0 μM of azidothymidine (AZT) (Moravek Biochemicals, Brea, CA, USA) was added to the culture medium the following day. The HIV-1 production in the culture medium was measured with an HIV-1 p24 antigen capture ELISA kit (ZeptoMetrix, Baffalo, NY, USA).

### 2.2. Isolation of DRM

The preparation of the DRM fractions from the Jurkat cells was described previously [22]. Briefly, the Jurkat cells (1 - 2 × 10^8^ cells) were lysed in 2.5 mL of MNE buffer (25 mM MES, 150 mM NaCl, 5 mM EDTA, 5 mM NaF, 1 mM sodium orthovanadate, and 1 mM PMSF, with pH 6.5) containing 0.2% Triton X-100 for 10 min on ice. They were homogenized with ten strokes in a Dounce homogenizer. The lysate was mixed with an equal volume of 80% (*w*/*v*) sucrose in MNE buffer and placed at the bottom of a Beckman SW41 ultracentrifuge tube. The sample was overlaid with 5 mL of 30% (*w*/*v*) sucrose in MNE buffer and 2 mL of 5% (*w*/*v*) sucrose in MNE buffer. The 40%–30%–5% sucrose step gradients were centrifuged at 125,000× *g* for 16 h at 4 °C in a Beckman SW41 rotor. The gradients were fractionated from top to bottom and analyzed using Western blotting. The DRM fractions were pooled, diluted with MNE buffer, and pelleted using ultracentrifugation at 125,000× *g* for 1 h. The DRM proteins were purified from the pellets with a 2D Clean-Up Kit (GE Healthcare, Chicago, IL, USA).

### 2.3. 2D DIGE and LC-MS

The HIV-1-infected Jurkat cells (1 × 10^8^ cells) were cocultured with the uninfected Jurkat cells (1 × 10^8^ cells) at 37 °C for 1 h. The DRM proteins (50 μg) purified from the cocultured samples with the 2D Clean-Up Kit were labeled with IC5-OSu fluorescent dye (DOJINDO Laboratories, Kumamoto, Japan) in a buffer containing 7 M urea, 2 M thiourea, 1% CHAPS, and 20 mM Tris-HCl (pH 8.5) for 30 min. The DRM proteins (50 μg) purified from the non-cocultured samples (the mixture of HIV-1-infected Jurkat and uninfected Jurkat cells without coculture, 1 × 10^8^ cells each) were similarly labeled with IC3-OSu fluorescent dye (DOJINDO Laboratories). The two labeled samples and unlabeled DRM proteins (200 μg each) were mixed and subjected to first-dimension isoelectric focusing on immobilized pH gradient strips (24 cm; pH 3–10 nonlinear) using an Ettan IPGphor II system (GE Healthcare). A second-dimension separation was performed on 9% SDS-PAGE (20 × 24 cm). The protein spots were visualized with a fluorescence scanner (Typhoon 9400, GE Healthcare), with 633 and 532 nm lasers for IC5-OSu and IC3-OSu. An image analysis was carried out with the DeCyder software (version 5.01, GE Healthcare). The proteins in the gel were transferred onto PVDF membrane and the protein spots excised were subjected to lysyl endopeptidase digestion. The resulting peptides were analyzed via nano-liquid chromatography-electrospray ionization mass spectrometry using a DiNa nano-LC system (KYA Technologies, Tokyo, Japan) coupled with a QStar Elite hybrid LC-MS/MS system (AB Sciex, Tokyo, Japan). A protein identification was carried out using the Protein Pilot version 3.0 software (AB Sciex) with the default parameters. The proteins identified with more than 99% (Protein Pilot Score > 2.0) were adopted.

### 2.4. Western Blotting

The protein samples were subjected to SDS-PAGE and transferred onto PVDF membrane. The membrane was incubated with primary antibodies and subsequently with secondary antibodies conjugated with HRP. The blots were developed with ECL systems and analyzed with Image Quant LAS500 (GE Healthcare). The following primary antibodies were used: mouse anti-HIV-1 p24CA [36], rabbit anti-Lck (Y123, Abcam, Cambridge, UK), mouse anti-CD71/transferrin receptor (TfR) (3B8 2A1, Santa Cruz Biotechnology, Dallas, TX, USA), mouse anti-Ly-GDP-dissociation inhibitor (G-12, Santa Cruz Biotechnology), rabbit anti-vimentin (EPR3776, Abcam), mouse anti-eukaryotic initiation factor 4A (eIF4A) (H-5, Santa Cruz Biotechnology), mouse anti-mitochondrial ATP synthase (51, Santa Cruz Biotechnology), anti-OxPhos Complex IV subunit I (COX) (1DC, Invitrogen, Thermo Fisher, Waltham, MA, USA), mouse anti-ERp57/protein disulfide isomerase (PDI) A3 (MaP.ERp57, Santa Cruz Biotechnology), mouse anti-vacuolar-type proton ATPase (F-6, Santa Cruz Biotechnology), mouse anti-tubulin (DM1A, Sigma-Aldrich, St. Louis, MO, USA), and mouse anti-actin (AC-15, Sigma-Aldrich) antibodies.

### 2.5. Confocal Microscopy

The HIV-1-infected Jurkat cells were incubated with the uninfected Jurkat cells that were prelabeled with 3.0 μM of Celltracker Orange CMTMR (Thermo Fisher Scientific, Waltham, MA, USA) at a 1:1 ratio (0.5 × 10^6^ cells each in 1 mL) on poly-L-lysine (PLL)-coated coverslips at 37 °C for 1 h. The cells were fixed with 2% paraformaldehyde (PFA) in phosphate-buffered saline (PBS) for 10 min, followed by permeabilization with 0.1% Triton X-100 for 10 min. After blocking with 1% FBS, the cells were incubated with primary antibodies (above) and subsequently with Alexa Fluor 488-conjugated anti-mouse or anti-rabbit IgG (Invitrogen, Thermo Fisher). For the DRM, the cells were immunostained with mouse anti-LAT1 (LAT1111, BioLegend, San Diego, CA, USA) and rabbit anti-caveolin-1 (ab2910, Abcam) antibodies. The cells were also immunostained for the tetraspanin-enriched membrane with mouse anti-CD63/LAMP3 antibodies (KILL150A, Santa Cruz Biotechnology). HIV-1 antigens were detected with mouse anti-HIV-1 p24CA antibodies [36]. The cell nuclei were stained with DAPI. The cells were observed with a laser scanning confocal microscope equipped with a ×60 oil immersion objective lens (TCS-SP5II, Leica Microsystems, Germany). The CMTMR-positive and -negative cell conjugates with broad binding interfaces (observed with differential interference contrast [DIC]) were subjected to a quantitative analysis of the antigen polarization. For polarization efficiency, the number of the cell conjugates displaying antigen accumulation to the cell interface was counted and their percentage in the total number of the cell conjugates used for the analysis was calculated.

For a single round of infection, vimentin-deficient Jurkat and CRISPR control Jurkat cells were infected with HIV-1 at an MOI of 3 and incubated at 37 °C for 2 days in the presence of 1.0 μM of AZT. The cells were immunostained with anti-HIV-1 p24CA antibodies. The percentage of p24CA-positive cells in the total cells was calculated. For cell conjugation, the cells were incubated on PLL-coated coverslips at 37 °C for 1 h. After immunostaininig, the cells were subjected to microscopy. The cell conjugation efficiency was shown as the percentages of the cell conjugates displaying broad binding interfaces (observed by DIC microscopy) in the total cells used for the analysis.

For HIV-1 VS formation, CRISPR control Jurkat cells (expressing Cas9 alone) were infected with HIV-1 at an MOI of 3 and incubated for 2 days. The cells were incubated with uninfected vimentin-deficient Jurkat cells at a 1:1 ratio (0.5 × 10^6^ cells each in 1 mL) at 37 °C for 1 h. Reversely, CRISPR control cells and vimentin-deficient cells were infected with HIV-1 at MOIs of 3 and 5, respectively. After 2 days of incubation, the cells were cocultured with uninfected control cells at a 1:1 ratio at 37 °C for 1 h. The cells were fixed, permeabilized, and immunostained with anti-HIV-1 p17MA antibodies specific to the C-terminal epitope of the mature p17MA (APR342, NIBSC, Hertfordshire, UK) [37]. The HIV-1 VS was defined as an accumulation of p17MA to the interface of the infected–uninfected cell conjugates. The number of cell conjugates displaying p17MA accumulation at the interface was counted and the percentage of these cell conjugates in the total infected–uninfected cell conjugates was calculated. For the immunostaining of the HIV-1 receptor/coreceptors, the cells were mixed at a 1:1 ratio and incubated in the presence of mouse anti-CD4 (OKT4, BD Pharmingen, Flanklin Lakes, NJ, USA) and anti-CXCR4 (12G5, BD Pharmingen) antibodies. After fixation with PFA, the cells were immunostained with human anti-HIV-1 gp120 antibodies (3501, ImmunoDX, Woburn, MA, USA) and subsequently with Alexa Fluor 488-conjugated anti-mouse IgG and Alexa Fluor 568-conjugated anti-human IgG (Invitrogen, Thermo Fisher). The number of HIV-1-infected (defined as gp120-positive) and uninfected cell conjugates displaying CD4/CXCR4 accumulation at the cell interface was counted. For this quantification, the percentages of the cell conjugates with antigen polarization in the total cell conjugates were calculated.

### 2.6. Flow Cytometry (FCM)

The cells were incubated with mouse anti-CD4 (OKT4), anti-CXCR4 (12G5), and anti-LFA-1α (TS2/4, BioLegend, San Diego, CA, USA) antibodies in PBS containing 1% FBS containing 0.1% NaN_3_ on ice for 30 min. The cells were subsequently incubated with Alexa Fluor 488-conjugated anti-mouse IgG (Invitrogen, Thermo Fisher, CA, USA) on ice for 30 min. After being fixed with 2% PFA in PBS, the cells were subjected to FCM with FACSMelody (BD, Flanklin Lakes, NJ, USA). A total of 10,000 events was processed for each sample.

## 3. Results

### 3.1. Accumulation of Lipid Raft Markers at HIV-1 VS

Previous studies have shown that lipid raft markers and tetraspanins accumulate at the cell interface of the HIV-1 VS [23,33]. To confirm these findings, Jurkat cells were infected with HIV-1 and cocultured with CMTMR-prelabeled Jurkat cells for 1 h. Immunostaining with anti-HIV-1 p24CA antibodies, followed by confocal microscopy, showed the accumulation of HIV-1 p24 at the interfaces of the infected and uninfected Jurkat cells, confirming VS formation (Figure 1). A confocal analysis also revealed the accumulation of the lipid raft marker LAT1, caveolin, and the tetraspanin CD63 at the cell–cell interface.

### 3.2. Proteomic Analysis of the DRM Fractions of Jurkat Cells Forming HIV-1 VS

Since lipid rafts are characterized by the DRM, the DRM was isolated from the Jurkat cells via treatment with cold Triton X-100, followed by membrane flotation centrifugation using a discontinuous sucrose density gradient (5–30–40%). Western blotting of the gradient fractions showed that most Lck (raft marker) was enriched in the DRM fractions, whereas that of TfR (non-raft marker) was distributed in the detergent-soluble membrane (DSM) fractions (Figure 2A). These results confirmed the enrichment of the lipid raft fractions using this isolation method.

To explore the molecules enriched in the HIV-1 VS, we cocultured HIV-1-infected and uninfected Jurkat cells with cell contact at 37 °C for 1 h. The DRM fractions were isolated from the cocultured cell samples and the DRM-associated proteins were labeled with the fluorescent dye IC5-OSu (Figure 2B,C, in red). For comparison, the infected and uninfected Jurkat cells (without cell contact) were collected using centrifugation and their DRM-associated proteins were labeled with IC3-OSu (Figure 2B,C, in green). The samples were mixed and subjected to 2D DIGE. For protein identification, 400 μg of unlabeled proteins from the cocultured cell samples was mixed together. Most protein spots displayed a yellow appearance due to the merging of the red and green fluorescence, indicating that the protein levels were equivalent in both samples. However, in the 2D DIGE images, a considerable number of red and green spots were observed, suggesting that the proteins accumulating in the DRM fractions under the cocultured and non-cocultured conditions were different (Figure 2C). Six independent experiments showed reproducible protein patterns comprising numerous differentially expressed protein spots. Some additional 2D DIGE images are shown in Appendix A. A quantitative analysis of the fluorescence intensity of each spot using the DeCyder program, as shown in Figure 2C, identified 132 spots that exhibited more than a 1.5-fold increase and 161 spots that exhibited more than a 1.5-fold decrease in the fluorescence intensity under the cocultured conditions in comparison to the non-cocultured conditions (Appendix A). We selected 20 spots that exhibited different expression levels between the two samples (in at least two experiments) and had a higher abundance (Table 1). These included 18 red and 2 green spots (Figure 2C). They were excised and subjected to protease digestion, followed by an LC-MS analysis (Table 1). Several other green and red spots were also observed; however, their expression levels were insufficient for an MS analysis.

### 3.3. Identification of Proteins Enriched in the DRM upon Formation of HIV-1 VS

The IC5-OSu-dominant (red) spots corresponded to the proteins enriched in the DRM upon cell contact, leading to VS formation. They were isolated from the 2D gels and subjected to LC-MS (Table 1). These were identified as ATP-related enzymes (mitochondrial ATP synthase subunit (mt-ATP synthase) and v-type proton ATPase subunit B (v-ATPase)), protein translation factors (eukaryotic initiation factor 4A (eIF4A) and mitochondrial elongation factor Tu (EF-Tu)), protein disulfide isomerase (PDI) A3, and 26S protease regulatory subunit 6A/6B. Charged multivesicular body protein 4B (CHMP-4B), an ESCRT-III molecule required for HIV-1 particle budding [38], was detected, confirming that CHMP-4B was recruited to the site of HIV-1 budding. GTP-related factors, Rho GDP-dissociation inhibitor 2, and GRIP1-associated protein 1 (guanine nucleotide exchange factor for the Ras GTPase family (RasGEF)) were also identified, although their incorporation into the DRM was low. Interestingly, three of the spots were identified as vimentin, with slight differences in their molecular weights and isoelectric points, possibly due to differential protein modifications. In contrast, the IC3-OSu-dominant (green) spots were similarly analyzed and found to include mitochondrial membrane protein (Metaxin-1) and nuclear membrane protein (lamin B1). Lipid raft markers, such as Lck/Fyn and GPI-anchored CD59, were not observed in the analyzed spots.

To confirm these findings, the cocultured/non-cocultured cell samples were treated with cold Triton X-100 in parallel and subjected to a membrane flotation analysis, followed by Western blotting (Figure 3 and Appendix A). We initially observed the distribution of Lck and found that, in both cell coculture/non-coculture samples, the majority of the Lck was distributed in the DRM fractions and its levels were very similar (72 ± 8% vs. 70 ± 5%), indicating that the total Lck levels associated with the DRM were largely unchanged in the cell contact condition. In contrast, vimentin was enriched in the DSM fractions in the non-cocultured sample (61 ± 6%) and in the DRM fractions in the cocultured sample (45 ± 7%), indicating that vimentin was incorporated into the DRM upon cell contact. eIF4A, PDI A3, and v-ATPase were originally enriched in the DSM fractions, and only small populations were distributed in the DRM (2 ± 1%, 2 ± 1%, and 6 ± 3%, respectively) under the non-cocultured conditions. However, their distributions in the DRM increased slightly upon cell contact (9 ± 3%, 7 ± 2%, and 25 ± 3%, respectively). The majority of mt-ATP synthase was associated with the DRM (65 ± 8%), but further incorporated into the DRM upon cell contact (98 ± 1%). These results were consistent with the data obtained via the 2D DIGE analysis. A literature search revealed that several of the molecules identified by our 2D DIGE analysis had been suggested to be involved in HIV-1 infection by previous studies (see below). The Rho GDP-dissociation inhibitor 2 was exclusively enriched in the DSM fractions of both cell samples (>99%), indicating that its incorporation into the DRM is essentially rare.

### 3.4. Polarized Localization of the Identified Molecules to HIV-1 VS

We explored whether the molecules that displayed increased incorporations into the DRM upon cell contact were indeed accumulated at the cell–cell contact sites. The HIV-1-infected Jurkat cells were cocultured with CMTMR-prelabeled Jurkat cells for 1 h. The cocultured cell samples were immunostained and subjected to confocal microscopy (Figure 4). The confocal microscopy showed that mt-ATP synthase was polarized toward the contact sites of the infected and uninfected cells. The mitochondrial marker COX2 was also polarized to the contact sites. These observations were consistent with those of previous studies, in which mt-ATP synthase accumulated at the contact sites between HIV-1-primed dendritic cells and target T cells [16], and mitochondria were polarized to the contact sites of HIV-1-infected and uninfected T cells [14,15]. When the polarization efficiency of these molecules to the VS was evaluated based on the percentage of antigen-positive cell conjugates in the infected–uninfected cell pairs, mt-ATP synthase and COX2 were found to be polarized in 34% and 32% of the cell conjugates, respectively. Our confocal study also showed that eIF4A, PDI A3, and v-ATPase were polarized to the cell-contact sites (49%, 33%, and 39% of the cell conjugates, respectively). PDI has been shown to reduce the disulfide bonds in HIV-1 Env, leading to its fusogenic conformation [39,40]. It is possible that PDI is recruited to the cell-contact sites to facilitate Env-mediated membrane fusion in the HIV-1 VS.

The HIV-1 Gag protein recruits the ESCRT machinery (ESCRT-0, I, II, and III) for particle budding [38]. The subsequent recruitment of VPS4 ATPase is required for the membrane scission process [41]. Our 2D DIGE analysis failed to detect VPS4, but did detect v-ATPase and CHMP-4B. v-ATPase is a multisubunit enzyme that hydrolyzes ATP and regulates vacuolar acidification. Although speculative, v-ATPase may also contribute to particle budding through ATP hydrolysis.

### 3.5. Vimentin Supports HIV-1 Infection

Previous studies have shown that the actin cytoskeleton is dramatically rearranged to form a SMAC-like structure upon VS formation [2,11]. Our 2D DIGE analysis revealed an increase in vimentin in the DRM fractions upon cell contact (Figure 2), and the confocal analysis showed that vimentin accumulated at the contact sites of the infected and uninfected cells (37%) (Figure 4). In this study, we explored the role of vimentin in HIV-1 infection. To this end, the Jurkat cells were transduced with lentivirus-expressing Cas9 and gRNA-targeting vimentin (knockout), or Cas9 alone (control). Vimentin knockout was confirmed using Western blotting (Figure 5A). The cell phenotypes, including the doubling times, were not affected by the vimentin knockout, as described previously [42]. These Jurkat lineages were infected with HIV-1 at a high MOI for 2 days and analyzed using Western blotting. In the vimentin-deficient cells, the expression level of the Gag protein was lower than that of the control cells (Figure 5B, left). The immunofluorescence staining for p24CA confirmed these findings (Figure 5B, right). These results suggest that vimentin supports HIV-1 infection. In accordance, HIV-1 production, as monitored by a p24 capture ELISA, was lower in the vimentin-deficient cells than the control cells (Figure 5C).

### 3.6. Little Effect of Vimentin Knockout on Cell Adhesion and Receptor Expression

HIV-1 infection was impaired in the vimentin-knockout cells (Figure 5). This raised the possibility that the expression of HIV-1 receptor/coreceptors on the cell surface was suppressed in the vimentin-deficient cells. To explore this possibility, the cell surface expression of CD4/CXCR4 was monitored using FCM. The results indicated that the expression level of CD4 was not significantly reduced by vimentin knockout (Figure 6A, left). The expression level of CXCR4 in the vimentin-knockout cells was comparable to that in the control cells (Figure 6A, right).

Recent studies have indicated that not only actin, but vimentin also play key roles in cell–cell contact by regulating the integrin function in endothelial and epithelial cells [43]. In endothelial cells, β1 and β3 integrin chains associate with vimentin to link the endothelial cells to the extracellular matrix [43,44,45]. In lymphocytes, LFA-1 (a heterodimer of αL and β2 chains) is a major surface molecule that regulates cell adhesion to form the immunological synapse and the VS [8,11,12]. It is possible that vimentin analogously contributes to the cell–cell contact in synapse-like structures through the LFA-1 β2 chain. We investigated the efficiency of this cell–cell contact in vimentin-deficient cells. The conjugation efficiency of the vimentin-deficient cells was equivalent to that of the control cells (Figure 6C). In accordance, the cell surface expression of LFA-1α, as monitored with FCM, was at similar levels in the vimentin-deficient and control cells (Figure 6B). These data suggest that vimentin is not responsible for cell–cell contact, at least in Jurkat cells.

### 3.7. Reduction in CD4 and CXCR4 Accumulation to HIV-1 VS in Vimentin-Deficient Cells

To better understand the role of vimentin in VS formation, control cells were infected with HIV-1 and cocultured with CMTMR-prelabeled vimentin-knockout cells. The cocultured cell samples were immunostained with anti-HIV-1 p17MA antibodies that only recognize the C-terminal epitope of mature p17MA exposed upon virus particle maturation [37]. The use of this type of p17MA antibody allowed us to detect the site of the particle budding [46,47]. Confocal microscopy revealed that the p17MA accumulated at the contact interface between the p17MA-positive and uninfected cells (Figure 7A, upper). To quantify the VS formation, approximately 200 infected–uninfected cell conjugates were observed, and the percentage of p17MA accumulation at the cell interface in the cell conjugates was calculated. The HIV-1 VS was more efficiently formed in the coculture with the control cells than in that with the vimentin-knockout cells (Figure 7A, lower), suggesting that the presence of vimentin in acceptor cells contributes to the efficient formation of the HIV-1 VS.

Next, we explored the role of vimentin in the donor cells. The vimentin-knockout and control cells were infected with HIV-1 and cocultured with the CMTMR-prelabeled control cells. After immunostaining for HIV-1 p17MA, VS formation was observed using confocal microscopy (Figure 7B, upper). A quantitative analysis revealed that the HIV-1 VS was efficiently formed no matter which cells were used as donors (Figure 7B, lower), indicating that the absence of vimentin in the donor cells did not affect the VS formation.

Our FCM data indicated that the cell surface expression of CD4/CXCR4 was not significantly reduced by the vimentin knockout (Figure 6A). Previous studies have shown that CD4/CXCR4 accumulates at the cell interface upon the formation of the HIV-1 VS [8]. We investigated whether this CD4/CXCR4 accumulation in the VS was impaired when the infected cells were conjugated with the vimentin-knockout cells. The cocultured cell samples were subjected to immunostaining for CD4/CXCR4 and HIV-1 gp120 (Figure 7C, upper). The accumulation efficiency of CD4/CXCR4 in the VS was evaluated based on the percentage of cell conjugates with CD4/CXCR4 accumulation in a population of HIV-1 gp120-positive cell conjugates. The accumulation, especially of CD4, was markedly impaired in the cell conjugates with vimentin-knockout cells (Figure 7C, lower), suggesting that the vimentin in the acceptor cells was responsible for the recruitment of CD4 to the contact site.

## 4. Discussion

Our 2D DIGE analysis showed that the contact between the HIV-1-infected and uninfected cells induced a differential protein accumulation in the DRM fractions. Compared to a recent proteomic analysis of whole cell lysates [34], our analysis used the DRM fractions and identified a much smaller number of proteins (Figure 2C). These included ATP synthase, CHMP-4B, and PDI, which have previously been suggested to be linked to HIV-1 VS formation, particle budding, and Env-mediated fusion, respectively [16,38,39]. However, the DRM incorporation of PDI upon cell contact was less evident, because the vast majority of PDI was present in the DSM. Interestingly, our analysis revealed that protein synthesis and quality-control-related factors, such as eIF4A, EF-Tu, and 26S protease subunits, which are normally cytosolic and not associated with the DRM, were enriched in the DRM fractions upon cell–cell contact. Although we cannot rule out contamination in the DRM fractions during preparation, it is tempting to speculate that polarized particle budding at the VS might be accompanied by the polarization of protein synthesis and trafficking machinery for a more efficient particle assembly at a certain time point. Another route in which a Gag cluster forms and moves on the plasma membrane toward the contact site has been reported [48].

We observed that mt-ATP synthase and mitochondrial COX2 were accumulated in the VS (Figure 3). Previous studies have made similar observations about mitochondrial polarization and suggested that they likely provide energy to the VS, where ATP is highly demanded [14,15,16]. However, we also found that the mitochondrial outer membrane protein (Metaxin-2) was reduced in the DRM fractions upon VS formation (Figure 2). One possibility is that ATP-related enzymes are pulled out of the mitochondrial membrane and recruited to the DRM fraction of the VS; however, other unnecessary enzymes are not recruited or excluded. This hypothesis is supported by a careful observation of the electron micrograph of the HIV-1 VS, in which some mitochondria underneath the VS were fragmented and partially damaged [15]. Alternatively, these protein segregations might simply be due to their differential affinities for the DRM constituents [49]. Our 2D DIGE analysis showed that lamin B1, a nuclear membrane protein, was reduced in the DRM fractions upon cell contact. The mechanism for this is unknown, although lamin B1 has been shown to be isolated in the DRM fraction [50]. Serum albumin, due to its high abundance in the cell culture, was likely to be accidentally incorporated into the DRM fraction.

It is well documented that both the immunological synapse and VS induce the dramatic remodeling of actin and tubulin networks [2,12,51,52,53,54]. At both synapses, a button-like actin network is formed at the contact site of the acceptor cells. MTOC are polarized in the acceptor cells in the immunological synapse and donor (infected) cells in the VS. A recent phosphoproteomic study of the HIV-1 VS revealed a significant activation of TCR signaling, followed by CD28 costimulation and actin cytoskeleton signaling [34]. Our 2D DIGE analysis of the DRM did not detect actin signaling, but found several forms of vimentin (Figure 2). In epithelial cells, intermediate filaments associate with integrins, which links the cells with the extracellular matrix through hemidesmosomes. In endothelial cells, vimentin is similarly linked to the extracellular matrix by binding integrins. Because these integrins reside in the DRM and are associated with actin networks, the vimentin-associated matrix adhesion (VAM) structure [43,44] possibly collaborates with actin signaling and/or acts to ensure its association with the extracellular matrix between cell conjugates. A direct interaction between actin and vimentin has also been reported [55].

Our study with vimentin-knockout cells showed that vimentin contributes to CD4 accumulation at contact sites, rather than HIV-1 particle budding (Figure 6). Vimentin has many phosphorylation sites and its functions are regulated by phosphorylation [56,57]. Numerous studies have shown that vimentin regulates the organization of the protein complexes on the cell membrane (e.g., recruitment of Src kinase) and acts as a scaffold for signaling molecules [58,59]. It is possible that the lateral movement of CD4 on the cell membrane to accumulate at the HIV-1 VS is facilitated by the presence of underlying vimentin. Without the local accumulation of CD4, HIV-1 infection may be insufficient, as observed in the HIV-1 infection of vimentin-deficient cells (Figure 5). However, because vimentin networks interact with various kinases, the contribution of vimentin to the inward signaling in acceptor cells cannot be ruled out. The involvement of vimentin in HIV-1 infection has been suggested by a previous study, in which vimentin knockdown suppressed HIV-1 replication in MT4 cells [42]. Interestingly, the study also showed that protein expression is reduced even when HIV-1 is pseudotyped with vesicular stomatitis virus G protein, which is suggestive of a postentry block of HIV-1 in vimentin-knockdown cells [42]. It is possible that the cell transmission of HIV-1 is further inhibited after viral entry. In summary, although phosphoproteomics is a powerful technique for studying cell-signaling pathways, a proteomic approach focusing on lipid raft/DRM fractions can provide additional perspectives for understanding the molecular mechanisms underlying HIV-1 VS formation.

## Figures and Tables

**Figure 1 viruses-15-01266-f001:**
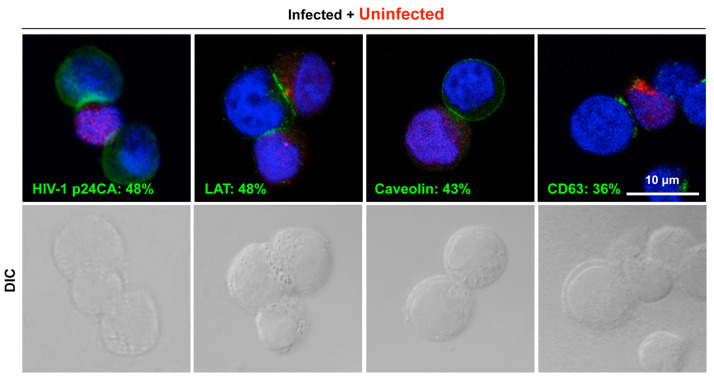
Accumulation of lipid raft markers and tetraspanins to the cell–cell contact sites in HIV-1 VS. Jurkat cells were infected with HIV-1 (the NL strain) at an MOI of 3. At 2 days post-infection (dpi), the cells (0.5 × 10^6^ cells) were incubated with Celltracker CMTMR-prelabeled uninfected Jurkat cells (0.5 × 10^6^ cells) with cell-to-cell contact at 37 °C for 1 h. The cell coculture samples were subjected to immunostaining with anti-HIV-1 p24CA, anti-LAT1, anti-caveolin (for lipid rafts), and anti-CD63 antibodies (for tetraspanins), and subsequently with Alexa Fluor 488-conjugated secondary antibodies. Cell nuclei were stained with DAPI. Representative images (a z-slice image) were shown at the same magnification. All were taken at the same magnification. Scale bar, 10 μm. The CMTMR-positive and -negative cell pairs were subjected to antigen localization analysis. The percentages of the cell pairs with antigen accumulation at the interface of cell conjugates in the total CMTMR-positive and -negative cell pairs used for analysis are shown.

**Figure 2 viruses-15-01266-f002:**
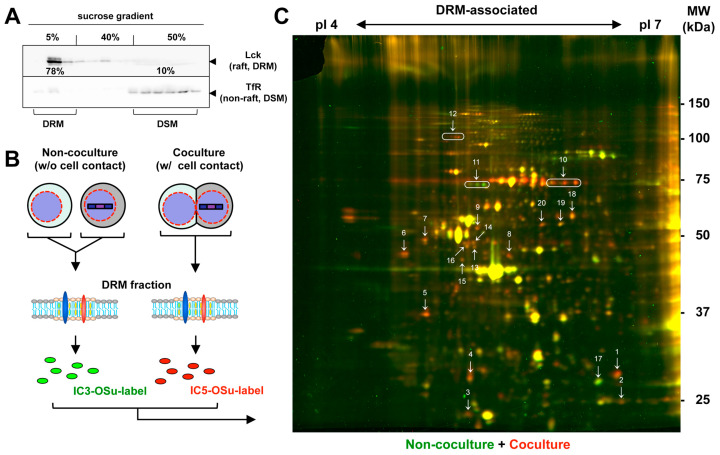
2D DIGE analysis of the DRM proteins isolated from coculture and non-coculture samples of HIV-1-infected and uninfected Jurkat cells. (**A**) Isolation of the DRM. Jurkat cells were lysed with Triton X-100 for 10 min on ice and subjected to membrane flotation centrifugation using 40–30–5% sucrose step gradients. Gradient fractions were analyzed using Western blotting with anti-Lck (for raft-associated) and anti-TfR antibodies (for non-raft-associated). (**B**) Experimental procedure. Jurkat cells (1 × 10^8^ cells) were infected with HIV-1 and incubated with uninfected Jurkat cells (1 × 10^8^ cells) at 37 °C for 1 h. The cell coculture was similarly lysed and subjected to membrane flotation centrifugation. The DRM fractions were collected, diluted, and pelleted by centrifugation. Proteins isolated from the fractions were labeled with IC5-OSu dye. In parallel, infected and uninfected Jurkat cells were mixed and immediately collected without coculture. The proteins isolated from the DRM fractions were labeled with IC3-OSu dye. The DRM sample pairs (corresponding to 50 μg protein each) were mixed and subjected to IEF (pI 3–10) followed by 9% SDS-PAGE. (**C**) 2D DIGE of coculture and non-coculture cell samples. Representative gel images (in 6 independent experiments) are shown. IC5-OSu-dominant and IC3-OSu-dominat spots were purified (No. 1 to 20) and subjected to LC-MS.

**Figure 3 viruses-15-01266-f003:**
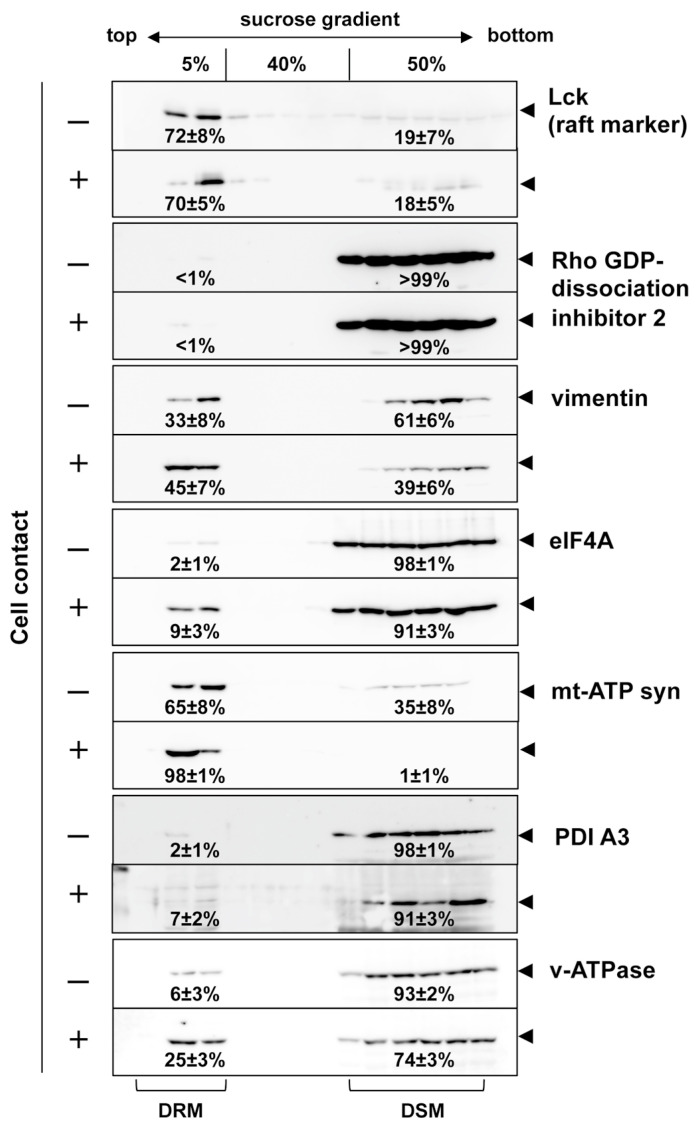
Incorporation of the identified molecules to the DRM fractions. Jurkat cells were infected with HIV-1 and incubated with uninfected Jurkat cells at 37 °C for 1 h, as described in the legend for Figure 2. The cell coculture was lysed and subjected to membrane flotation centrifugation. Gradient fractions were analyzed using Western blotting with anti-Lck, anti-Ly-GDP-dissociation inhibitor, anti-vimentin, anti-eIF4A, anti-mt-ATP synthase, anti-ERp57/PDI A3, and anti-v-ATPase antibodies. Representative blots were shown. The percentage of the protein distribution in the DRM and DSM fractions was quantified with ImageJ software. Data are the mean ± SD from two or three independent experiments.

**Figure 4 viruses-15-01266-f004:**
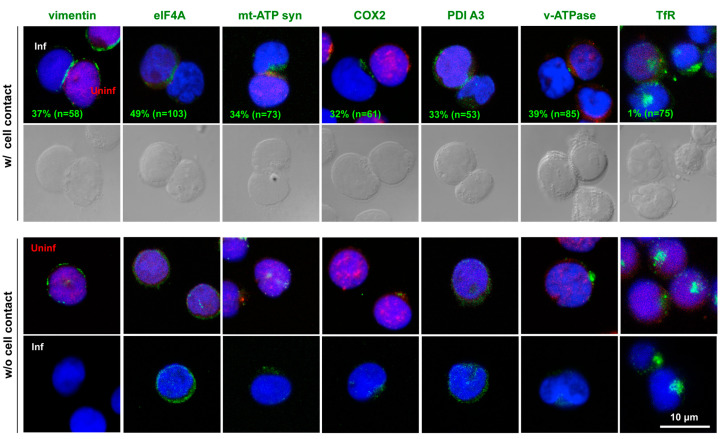
Polarization of the identified molecules to the cell–cell interfaces in HIV-1 VS. Jurkat cells were infected with HIV-1 and incubated with CMTMR-prelabeled uninfected Jurkat cells at 37 °C for 1 h, as described in the legend for Figure 1. The cells were immunostained with anti-vimentin, anti-eIF4A, anti-mt-ATP synthase, anti-COX2, anti-ERp57/PDI A3, anti-v-ATPase, and anti-TfR antibodies. Cell nuclei were stained with DAPI. Uninfected Jurkat and infected Jurkat cells (without coculture) were similarly immunostained. Representative images (a z-slice image) were shown at the same magnification. All were taken at the same magnification. Scale bar, 10 μm. The CMTMR-positive and -negative cell pairs were subjected to antigen localization analysis. For polarization efficiency, the percentages of the cell pairs with antigen accumulation to the interface of cell conjugates in the total CMTMR-positive and -negative cell pairs (the means from several experiments) are calculated. n indicates the numbers of the cell conjugates subjected to this analysis.

**Figure 5 viruses-15-01266-f005:**
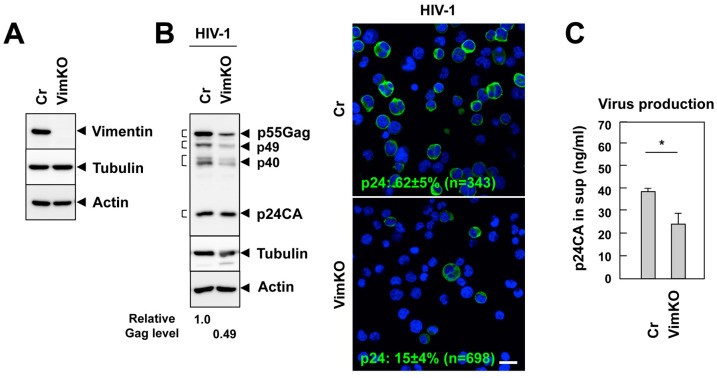
Vimentin supports HIV-1 infection. (**A**) Vimentin knockout and CRISPR control Jurkat cells were subjected to Western blotting with anti-vimentin, anti-tubulin, and anti-actin antibodies. (**B**,**C**) The Jurkat lineages were infected with HIV-1. The following day, AZT was added to avoid multiple rounds of HIV-1 replication. (**B**) At 2 days post-infection, the cells were subjected to Western blotting with anti-HIV-1 p24CA, anti-tubulin, and anti-actin antibodies (left). The band intensities of Gags (p55+p49+p40+p24) were quantified with ImageJ software and the relative expression level of total Gag is shown. The cells were immunostained with anti-HIV-1 p24CA antibodies (right). The ratios of p24CA-positive cells to the total cells were shown. n indicates the numbers of the cells subjected to this analysis. Representative images (a z-slice image) are shown. (**C**) Production of p24CA antigens in the culture medium was quantified with p24CA capture ELISA. Data are the mean ± SD from 3 independent experiments. * *p* < 0.05, Mann–Whitney U test.

**Figure 6 viruses-15-01266-f006:**
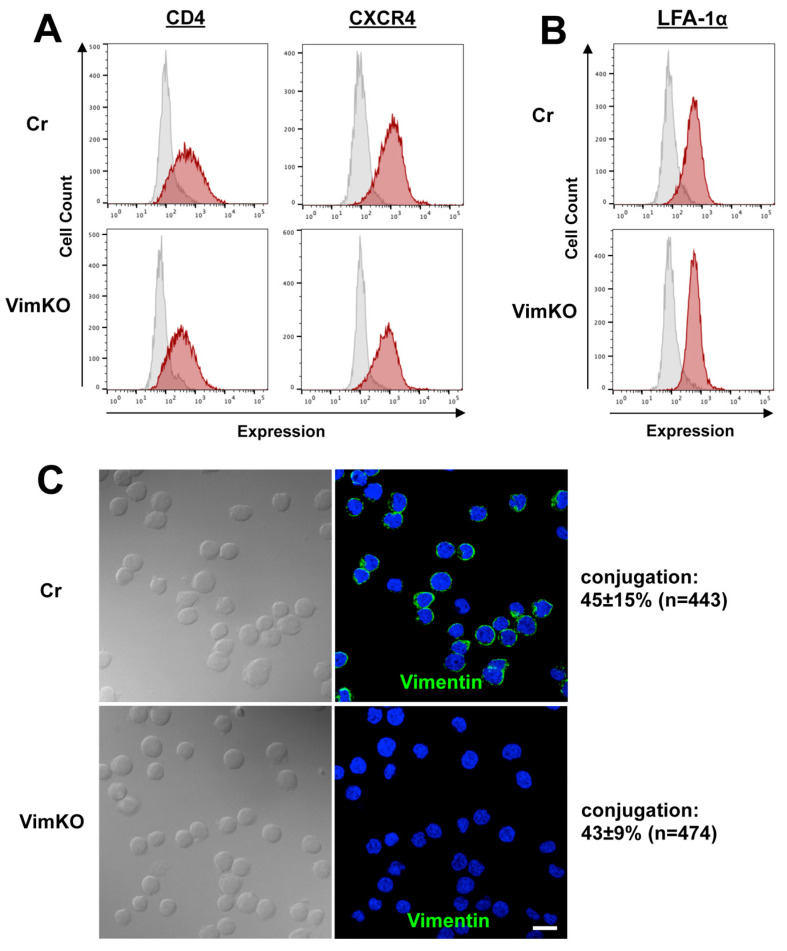
Vimentin knockout does not impair receptor expression and cell conjugation. Jurkat cells were transduced with lentivirus-expressing Cas9 alone (control) or Cas9 and gRNA-targeting vimentin (knockout). (**A**) Cell surface expression of HIV-1 receptors. Vimentin knockout cells and CRISPR control cells were immunostained with anti-CD4 and anti-CXCR4 antibodies and subjected to FCM. (**B**) Cell surface expression of LFA-1α. The cells were immunostained with anti-LFA-1α antibodies and subjected to FCM. Gray-shaded histograms show samples immunostained with secondary antibodies alone. Representative images are shown. (**C**) Cell conjugation efficiency. Vimentin knockout cells (1 × 10^6^ cells) were incubated at 37 °C for 1 h to allow cell conjugation. The control cells were similarly incubated at 37 °C for 1 h. The cells were immunostained with anti-vimentin antibodies. Cell nuclei were stained with DAPI. Representative images (a z-slice image) are shown. The cell conjugation efficiency was calculated as the percentage of cell conjugates with a cell-binding interface in total cells used for analysis. Data are the mean ± SD from three independent experiments.

**Figure 7 viruses-15-01266-f007:**
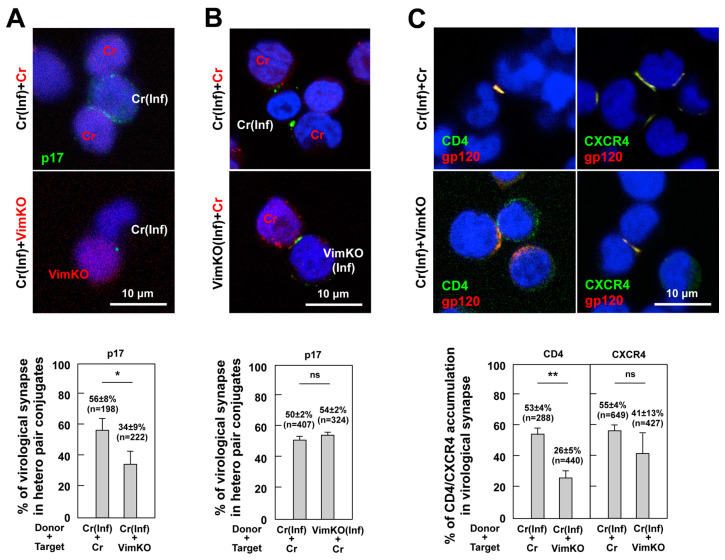
Formation of HIV-1 VS in vimentin-knockout Jurkat cells. (**A**) Formation of HIV-1 VS of vimentin-knockout donor cells. CRISPR control cells were infected with HIV-1 for 2 days. The cells were incubated with CMTMR-prelabeled, vimentin-knockout, or control cells on coverslips at 37 °C for 1 h, as described in the legend for Figure 1. The cell coculture samples were subjected to immunostaining with anti-HIV-1 p17MA antibodies specific to the mature p17MA but not to p55Gag. Cell nuclei were stained with DAPI. The CMTMR-positive and -negative cell conjugates were subjected to p17MA localization analysis. HIV-1 VS was defined as accumulation of HIV-1 antigens at the interface of cell conjugates. For quantification, the number of the cell conjugates with p17MA accumulation at the cell interface was counted and the percentages of the cell conjugates with the p17MA polarization in the total cell conjugates was calculated. (**B**) Formation of HIV-1 VS of vimentin-knockout acceptor cells. CRISPR control and vimentin-knockout cells were infected with HIV-1 for 2 days. The cells were incubated with CMTMR-prelabeled control cells at 37 °C for 1 h. The cell coculture samples were similarly subjected to immunostaining and confocal microscopy. (**C**) Accumulation of HIV-1 receptor/coreceptor in vimentin knockout cells. CRISPR control cells were infected with HIV-1 for 2 days. The cells were incubated with vimentin-knockout or control cells on coverslips in the presence of anti-CD4 and anti-CXCR4 antibodies at 37 °C for 1 h. After fixation of PFA, the cells were immunostained with anti-HIV-1 gp120 antibodies (without permeabilization). The accumulation efficiency of HIV-1 receptor/coreceptors at the cell–cell interface was shown as the percentages of CD4/CXCR4 at the cell interfaces in HIV-1-infected (defined as gp120-positive) and uninfected cell conjugates in the cell conjugates. Representative images (a z-slice image) were shown at the same magnification. Scale bar, 10 μm. The percentages are the mean ± SD from three independent experiments. ** *p* < 0.01; * *p* < 0.05; ns, not significant, Mann–Whitney U test. n indicates the numbers of the cell conjugates used for each analysis.

**Table 1 viruses-15-01266-t001:** Summary of the proteins analyzed using 2D DIGE.

Spot No.	Name	Accession No.	MW (KDa)	Spot Volume Ratio *
1	HIV-1 p24CA	NP579880.1	25.6	2.45
2	Unidentified			1.49
3	Mitochondrial ATP synthase subunit d	O75947	18.5	1.66
4	Rho GDP-dissociation inhibitor 2	P52566	23	1.76
5	Charged multivesicular body protein 4B	Q9H444	25	2.46
6	Vimentin	P08670	53.7	2.66
7	Vimentin	P08670	53.7	1.74
8	Eukaryotic initiation factor 4A	P06842	46.2	2.71
9	Vimentin	P08670	53.7	2.04
10	Serum albumin	P02768	69.4	3.19, 3.49, 3.53
11	Lamin B1	P20700	66.4	−1.11, −2.27, −3.07
12	GRIP1-associated protein 1	Q4V328	96	1.93, 2.50, 2.39, 3.24
13	Mitochondrial Elongation factor Tu	P494111	49.5	1.37
14	26S protease regulatory subunit 6A	P17980	49.2	1.5
15	unindentified			1.61
16	26S protease regulatory subunit 6B	P43686	47.4	1.61
17	Metaxin 2	O75431	29.8	−1.84
18	Protein disulfide-isomerase A3	P30101	56.8	1.58
19	V-type proton ATPase subunit B	P21281	56.5	1.16
20	V-type proton ATPase subunit B	P21281	56.5	1.8

* Spot volume ratio indicates the fold difference of spot volume between coculture and non-coculture conditions for each spot. Spot volume ratios for multiple spots are shown starting from the spot with lower pI, which is on the acidic side. Proteins identified with more than 99% (Protein Pilot Score > 2.0) were adopted.

## Data Availability

The raw data supporting the conclusions of this manuscript will be available by the authors on request.

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
