# Peer review of "A Proteomic Analysis of Detergent-Resistant Membranes in HIV Virological Synapse: The Involvement of Vimentin in CD4 Polarization"

_viruses, 2023, doi:10.3390/v15061266_

Round 1

Reviewer 1 Report

In this manuscript, the authors investigate the composition of detergent resistant membranes in HIV-1 infected T-cell cultures in the presence and absence of virological synapse formation using a 2D DIGE approach, supported by membrane flotation and immunofluorescence, with the aim to identify host cell proteins recruited to VS contact sites. Besides factors for which this recruitment has been shown or implicated previously, their main observation ist hat this includes vimentin, which they report to be involved in accumulation of CD4 at the VS region in the target cell. Overall the manuscript is well written and the results are in my view worth reporting to the readers of this journal as a starting point for further investigation.

My main concerns however refer to some limitations of the approach, which should be acknowledged (pints 1-3) and to the difficulty to assess the reprocibility of the findings based on the data provided (points 4ff).

1.      The term ‚raft‘ is used synonymously with DRM throughout the text. The terms were used interchangeably at earlier times, but I believe today the situation is viewed to be somewhat more complex. The low temperature used for extraction and the choice of detergent can influence DRM composition, and how this relates to the dynamic membrane microdomains in living cells is not fully understood. In the description of the findings, the statements refer excusively to DRM obtained under certain experimental conditions. Please acknowledge this point and change to DRM throughout.

2.      Fig. 2: The term Proteomic evokes a much more comprehensive approach. It is not clear whether the selection of 22 proteins in table 1 reflects a representative picture of the most prominent factors. I suggest to change the legend title to 2D DIGE analysis and briefly comment in the manuscript on why this appoach was chosen.

3.      With the exception of vimentin, the authors mainly focus on proteins which are in line with previous findings using other appoaches or cell types. However, Table 1 comprises also examples that one would not expect to be enriched or depleted at a VS and that are not mentioned in the text (2 out of 22, BSA, Lamin B1); the corresponding spots in Fig. 1 C visually appear convincing for selective enrichment. Please comment on how this might limit the appoach / interpretation of the findings.

4.      The 2D DIGE experiment was performed 5 (line 243) or 6 (line 254) times (??), but only a single experiment is shown in Fig. 2C and Fig. S1, and the numbers provided in line 259/260 presumably also refer to this one experiment. The word „repeatedly“ used in line 458 referring to this figure is thus not justified based on the data shown. In order to demonstrate reproducibility, data from additional experiments need to be shown in the supplement.

5.      From looking at Fig. 2C it is not obvious to me why the  boxed spots were chosen, while others were not (it might be visually helpful to display the two channels separately in black and white in the appendix). Please state the objective criteria underlying the selection. Were these criteria met in multiple experiments for spots detected at equivalent positions?

6.       Does the MS analysis reflect a single determination from one experiment? Line 353: what does ‚frequently‘ mean with respect to this table?

7.      Figure 3: Again, data from one experiment are shown. Please provide additional data from other replicates in the supplement. ECL-based immunoblot is not a very quantitative approach and depends very much on the conditions and the antigen (I don’t think it is possible to discriminate between 2 and 5%), so it would be helpful to show that the observed differences are qualitatively reproducible. 

8.      Figure 5: knockout of vimentin might also affect cell viability and thereby HIV-1 expression. Was this controlled for?

9.      Legend to Fig. 4 is copy-paste from Fig. 3. Legend 5 is completely missing.

10.   The observations that vimentin knockdown impairs HIV-1 replication in T cells has been reported by others (https://doi.org/10.3390/v8060098; I am not sure if this ist he only report). I believe such literature should be cited and discussed.

Minor points

Fig. 1 shows examples of cells engaged in cell-cell contacts. Please provide an estimate of the proportion of cells in the sample engaged in such contacts under the conditions used at the time of harvest.

Fig. 3: please provide DIG images for lower panel as well.

Figure 6: how is a conjugate defined here?

Line 318: the molecules which

Line329 ff.: which n numbers do these percentges refer to?

Liene 404: particle maturation

Line 429: with PFA

Line 455: protein accumulation

Author Response

(Comments and suggestions by Reviewer 1)

In this manuscript, the authors investigate the composition of detergent resistant membranes in HIV-1 infected T-cell cultures in the presence and absence of virological synapse formation using a 2D DIGE approach, supported by membrane flotation and immunofluorescence, with the aim to identify host cell proteins recruited to VS contact sites. Besides factors for which this recruitment has been shown or implicated previously, their main observation ist hat this includes vimentin, which they report to be involved in accumulation of CD4 at the VS region in the target cell. Overall the manuscript is well written and the results are in my view worth reporting to the readers of this journal as a starting point for further investigation.

(Responses)

We thank the reviewer for considering that our study is worth reporting.

The reviewer has some concerns about the reproducibility of 2-D DIGE analysis. However, 2-D DIGE, although is gel-based analysis, is considered accurate and reproducible, as long as you pay attention to protein solubilization (e.g., DRM preparation in this study) and dye labeling (Proteomics Clin. Appl. 2015, 00, 1–12). We have now cited this review article in the text. We routinely perform 2-D DIGE analysis using various samples including lipid rafts (Nat Struct Mol Biol, 2009, 16, 1026-1035; Electrophoresis, 2007, 28, 2035-2043; Electrophoresis, 2014, 35,554-562) and are familiar with this technology.

Reviewer 1 indicated the 10 points with major concerns (below) and asked for improvement.

(Q1) The term ‚raft‘ is used synonymously with DRM throughout the text. The terms were used interchangeably at earlier times, but I believe today the situation is viewed to be somewhat more complex. The low temperature used for extraction and the choice of detergent can influence DRM composition, and how this relates to the dynamic membrane microdomains in living cells is not fully understood. In the description of the findings, the statements refer exclusively to DRM obtained under certain experimental conditions. Please acknowledge this point and change to DRM throughout.

(A1) As per the suggestion, most of the term “raft” in the manuscript have been replaced with “DRM”. However, the term “raft” in old literature and the terms “raft/non-raft markers” have been left unchanged.

(Q2) Fig. 2: The term Proteomic evokes a much more comprehensive approach. It is not clear whether the selection of 22 proteins in table 1 reflects a representative picture of the most prominent factors. I suggest to change the legend title to 2D DIGE analysis and briefly comment in the manuscript on why this approach was chosen.

(A2) In this study, we compared the DRM-associated proteins between the coculture and non-coculture cell samples in 2-D DIGE. We selected the spots from the following points: the differential fluorescent intensity between the two samples and the protein levels sufficient for identification by LC-MS. Therefore, low protein levels, even when the spots show significant differences in fluorescence, do not allow protein identification. When the spots were selected from 2-D DIGE, it was uncertain whether they were important for the VS formation. Then we further explored in Figs. 3-7.

As per the suggestion, the legend title for Fig. 2 has been changed to “2-D DIGE analysis ---“. Also, we have revised our 2-D DIGE data and have now limited to 20 spots, based on their reproducibility.

(Q3) With the exception of vimentin, the authors mainly focus on proteins which are in line with previous findings using other approaches or cell types. However, Table 1 comprises also examples that one would not expect to be enriched or depleted at a VS and that are not mentioned in the text (2 out of 22, BSA, Lamin B1); the corresponding spots in Fig. 2C visually appear convincing for selective enrichment. Please comment on how this might limit the approach / interpretation of the findings.

(A3) BSA: Because BSA is abundant in the cell culture, unwanted contamination to the DRM fraction cannot be rule out. We have added this comment to the discussion section.

Lamin B1: It has been shown to be incorporated to the DRM (Proteomics, 2009, 9, 3022-3035). We have now cited the paper.

(Q4) The 2D DIGE experiment was performed 5 (line 243) or 6 (line 254) times (??), but only a single experiment is shown in Fig. 2C and Fig. S1, and the numbers provided in line 259/260 presumably also refer to this one experiment. The word „repeatedly“ used in line 458 referring to this figure is thus not justified based on the data shown. In order to demonstrate reproducibility, data from additional experiments need to be shown in the supplement.

(A4) We performed 2-D DIGE analysis 6 times. Three additional 2-D DIGE images have now been shown in new Fig. S1. The previous Fig. S1 have been renamed to Fig. S2. We stated in the text that Fig. S2 was produced from Fig. 2C.

The sentence including the word “repeatedly” was eliminated.

(Q5) From looking at Fig. 2C it is not obvious to me why the boxed spots were chosen, while others were not (it might be visually helpful to display the two channels separately in black and white in the appendix). Please state the objective criteria underlying the selection. Were these criteria met in multiple experiments for spots detected at equivalent positions?

(A5) As stated in (A2), the spots were selected, based on the differential fluorescent intensity between the two samples (in at least 2 experiments) and the protein levels sufficient for MS analysis. We have now stated these points in the text.

(Q6) Does the MS analysis reflect a single determination from one experiment? Line 353: what does ‚frequently‘ mean with respect to this table?

(A6) Since our 2-D DIGE patterns, at least the positions of the spots, were reproducible, we performed the MS analysis for spot identification once or twice. Therefore, we eliminated the word “frequently” from the sentence.

(Q7) Figure 3: Again, data from one experiment are shown. Please provide additional data from other replicates in the supplement. ECL-based immunoblot is not a very quantitative approach and depends very much on the conditions and the antigen (I don’t think it is possible to discriminate between 2 and 5%), so it would be helpful to show that the observed differences are qualitatively reproducible. 

(A7) We have shown some additional sets of western images (membrane flotation data) in the supplementary figures (Fig. S3). Representative images are shown with the means of % distribution of the molecules and SD in the text.

(Q8) Figure 5: knockout of vimentin might also affect cell viability and thereby HIV-1 expression. Was this controlled for?

(A8) We generated Jurkat cells expressing Cas9 alone (without vimentin gRNA) and used as control. We have added a short text stating that knockout of vimentin did not affect cell phenotypes such as viability and have cited a paper reporting similar observation (viruses, 2016, 8, 98).

(Q9) Legend to Fig. 4 is copy-paste from Fig. 3. Legend 5 is completely missing.

(A9) We apologize for these duplication and omission. We have added correct legends for Figs. 4 and 5.

(Q10) The observations that vimentin knockdown impairs HIV-1 replication in T cells has been reported by others (https://doi.org/10.3390/v8060098; I am not sure if this ist he only report). I believe such literature should be cited and discussed.

(A10) We greatly thank for this suggestion. We have now cited this paper and added a discussion about the possibility of postentry block. We apologize the omission of this reference in our initial submission.

Minor points

(Q1) Fig. 1 shows examples of cells engaged in cell-cell contacts. Please provide an estimate of the proportion of cells in the sample engaged in such contacts under the conditions used at the time of harvest.

(A1) In Fig. 1, we added the polarization efficiencies of the antigens to the interface of infected-uninfected cell conjugates.

(Q2) Fig. 3: please provide DIG images for lower panel as well.

(A2) DIC microscopy was used to confirm the tight cell-to-cell contact, as if two cells were stuck to each other. Since the lower panels of Fig. 3 were regarding the cells without contact coculture, we think that their DIC images were unnecessary.

(Q3) Figure 6: how is a conjugate defined here?

(A3) We observed the cells by DIC microscopy. The cell conjugation was defined as having a broad binding interface between cell conjugates, as shown in Fig. 1.

(Q4) Line 318: the molecules which

(A4) The word “which” was inserted.

(A5) Line329 ff.: which n numbers do these percentages refer to?

(Q5) The “n” numbers shown in the legends refer to the numbers of cells or cell conjugates used for analysis.

(Q6) Line 404: particle maturation

Line 429: with PFA

Line 455: protein accumulation

(A6) We thank for the English editing and followed the suggestions.

Reviewer 2 Report

This paper contributes to the understanding of the molecular mechanisms involved in the HIV-1 T-cell virological synapse. It clearly shows an involvement of Vimentin in the accumulation of the CD4 receptor in lipid rafts thanks to the clever use of an elegant two-dimensional fluorescent difference gel electrophoresis approach. The paper is pleasant to read and present well the experimental logic and conclusions.

However more experimental details, as such in the material and methods section, would be necessary to provide more clarity to the presented experiments und results. 

  • There is a little typo in line 71 in the introduction: HV instead of HIV
  • Legend for figure 4 is completely wrong except maybe its title.
  • Legend for figure 5 is missing

The most necessary clarification concern the material and methods section:

  • In 2.1 "cell culture and viral infections" at line 94, there is only a mention of "the NL strain". Please indicate which one as well as how the virus particles were produced and isolated. 
  • In 2.1 "cell culture and viral infections" at line 104 to 105 a VSV pseudotyped strain is mentioned for infection of Jurkat cells. At no point this viruses appear to be used in the presented study nor would its use without a produced HIV env be useful in studies for the virological synapse. 
  • In 2.4 "western blot" the crucial details such as clone name or manufacturers reference to the antibodies used are missing: anti-Lck (line 151), anti-Vimentin (line 154), anti-mitochondrial ATP synthase (line 156), anti-ERp57 (line 158).
  • In 2.5 "confocal microscopy" the crucial details such as clone name or manufacturers reference to the antibodies used are missing: anti-caveolin (line 170), anti-CD63 (line 172).
  • In 2.5 "confocal microscopy" details to image analysis are missing. Experiments described in figure 4 (results section 3.4) mention the "percentage of antigen-positive cell conjugates in the infected-uninfected cell pairs". How were the cells pairs identified i.e. how was a virological synapse event distinguished from cells which happen to be touching after fixation and mounting of the sample for microscopy ? How was then the percentage of signal quantified? On one z slice image or a maximum intensity or other projection of a z-image stack?  The lens used for imaging should be indicated.
  • The quantification method of the microscopy experiments described in figure 5B (section 3.5) is absent from the material and methods section 2.5. How was the signal quantified and what kind of images were used (single slice or z projection of what kind?). Again, the lens used for imaging should be indicated.
  • The  methods for quantification of cell conjugation efficiency shown in figure 6 (section 3.6) are missing in section 2.5. How was a virological synapse event distinguished from cells which happen to be touching after fixation and mounting of the sample for microscopy ? What kind of images are shown here: single z slice or projection?
  • In methods for quantification of antigen accumulation at the cell to cell interface used in figure 7 are not described in section 2.5. How are the virological synapse events identified and subsequently how were the mentioned "ratios of p17MA accumulation" quantified? Again, what kind of images were used and shown in the figure: individual z-slices or projections and if so what kind?

Author Response

(Comments and suggestions by Reviewer 2)

This paper contributes to the understanding of the molecular mechanisms involved in the HIV-1 T-cell virological synapse. It clearly shows an involvement of Vimentin in the accumulation of the CD4 receptor in lipid rafts thanks to the clever use of an elegant two-dimensional fluorescent difference gel electrophoresis approach. The paper is pleasant to read and present well the experimental logic and conclusions.

However more experimental details, as such in the material and methods section, would be necessary to provide more clarity to the presented experiments und results. 

There is a little typo in line 71 in the introduction: HV instead of HIV

Legend for figure 4 is completely wrong except maybe its title.

Legend for figure 5 is missing

(Responses)

We thank the reviewer for appreciating our findings (vimentin-CD4 recruitment). We admit a small description of the materials and methods, and have added more information to the M&M section and the individual legends for figures.

We sincerely apologize for the errors in the legends for Figs. 4 and 5. We replaced the legend for Fig. 4 and added the legend for Fig. 5.

The most necessary clarification concern the material and methods section:

(Q1) In 2.1 "cell culture and viral infections" at line 94, there is only a mention of "the NL strain". Please indicate which one as well as how the virus particles were produced and isolated.

(A1) We have added some text on production of HIV-1. The details of an HIV-1 plasmid, pNL4-3 (AF324493.2) and the cells used for transfection have now been stated in this subsection.

(Q2) In 2.1 "cell culture and viral infections" at line 104 to 105 a VSV pseudotyped strain is mentioned for infection of Jurkat cells. At no point this viruses appear to be used in the presented study nor would its use without a produced HIV env be useful in studies for the virological synapse.

(A2) We apologize for mistakenly including this text in the subsection. We eliminated the text regarding VSV-G-pseudotyped HIV-1.

(Q3) In 2.4 "western blot" the crucial details such as clone name or manufacturers reference to the antibodies used are missing: anti-Lck (line 151), anti-Vimentin (line 154), anti-mitochondrial ATP synthase (line 156), anti-ERp57 (line 158).

(A3) The clone names of the antibodies have been added to the 2.4 subsection.

(Q4) In 2.5 "confocal microscopy" the crucial details such as clone name or manufacturers reference to the antibodies used are missing: anti-caveolin (line 170), anti-CD63 (line 172).

(A4) The clone names of the antibodies have been added to the 2.5 subsection.

(Q5) In 2.5 "confocal microscopy" details to image analysis are missing. Experiments described in figure 4 (results section 3.4) mention the "percentage of antigen-positive cell conjugates in the infected-uninfected cell pairs". How were the cells pairs identified i.e. how was a virological synapse event distinguished from cells which happen to be touching after fixation and mounting of the sample for microscopy ? How was then the percentage of signal quantified? On one z slice image or a maximum intensity or other projection of a z-image stack?  The lens used for imaging should be indicated.

(A5) Jurkat cells were infected with HIV-1 at a high MOI and cocultured with CMTMR-labeled uninfected Jurkat cells. The infected-uninfected cell pairs were identified as CMTMR-positive and negative cell pairs. Since the VS displays a broad binding interface between cell pairs (as if two cells are stuck to each other), the cell-to-cell conjugates representing the VS were confirmed by DIC microscopy (see the DIC images in the upper panel sets in Fig. 4). The DIC observation allows us to exclude just accidental cell attachment and cell aggregation which may occur during sample preparation. For quantification, we counted the number of the cell pairs showing accumulation of the antigens at the cell-cell interface. The polarization efficiency was shown as the percentages of the cell pairs with polarized antigen distribution to the interface in the total cell conjugates. We did not quantify the signal intensity in individual cell pairs. The images shown in this manuscript are one z-slice image throughout. We added some text to the 2.5 subsection and the legends for Figs. 1 and 4. The lens used for observation was stated in the subsection.

(Q6) The quantification method of the microscopy experiments described in figure 5B (section 3.5) is absent from the material and methods section 2.5. How was the signal quantified and what kind of images were used (single slice or z projection of what kind?). Again, the lens used for imaging should be indicated.

(A6) We have added some text for the quantification method to the 2.5 subsection and the legend for Fig. 5.

(Q7) The methods for quantification of cell conjugation efficiency shown in figure 6 (section 3.6) are missing in section 2.5. How was a virological synapse event distinguished from cells which happen to be touching after fixation and mounting of the sample for microscopy ? What kind of images are shown here: single z slice or projection?

(A7) We added some text (for quantification of cell conjugation) to the 2.5 subsection and the legend for Fig. 6.

(Q8) In methods for quantification of antigen accumulation at the cell to cell interface used in figure 7 are not described in section 2.5. How are the virological synapse events identified and subsequently how were the mentioned "ratios of p17MA accumulation" quantified? Again, what kind of images were used and shown in the figure: individual z-slices or projections and if so what kind?

(A8) We have added the methods for quantification of p17MA and CD4/CXCR4 accumulation at the cell interface in the 2.5 subsection. We also described the methods in the legend for Fig. 7.

Round 2

Reviewer 1 Report

The authors have addressed the points raised in my review.